# Enhancing photovoltages at p-type semiconductors through a redox-active metal-organic framework surface coating

Anna M. Beiler[1], Brian D. McCarthy [1], Ben A. Johnson [1] & Sascha Ott [1]✉

Surface modification of semiconductors can improve photoelectrochemical performance by promoting efficient interfacial charge transfer. We show that metal-organic frameworks (MOFs) are viable surface coatings for enhancing cathodic photovoltages. Under 1-sun illumination, no photovoltage is observed for p-type Si(111) functionalized with a naphthalene diimide derivative until the monolayer is expanded in three dimensions in a MOF. The surface-grown MOF thin film at Si promotes reduction of the molecular linkers at formal potentials >300 mV positive of their thermodynamic potentials. The photocurrent is governed by charge diffusion through the film, and the MOF film is sufficiently conductive to power reductive transformations. When grown on GaP(100), the reductions of the MOF linkers are shifted anodically by >700 mV compared to those of the same MOF on conductive substrates. This photovoltage, among the highest reported for GaP in photoelectrochemical applications, illustrates the power of MOF films to enhance photocathodic operation.

[1] Department of Chemistry, Ångström Laboratory, Uppsala University, Box 523, 75120 Uppsala, Sweden. ✉email: sascha.ott@kemi.uu.se

Metal–organic frameworks (MOFs) provide opportunities to build well-ordered three-dimensional structures for the generation of sustainable fuels and chemicals[1–4]. They combine the advantages of molecular species, with well-defined and precisely tunable active sites, with those of heterogeneous structures, which offer long-term stability and technological readiness. In addition, the porosity and extended architecture of MOFs give access to a high concentration of active sites per surface area. We recently reported a porous interpenetrated zirconium organic framework (PIZOF), a UiO-type MOF (UiO = University of Oslo) built from hexanuclear zirconium oxide secondary building units (SBUs) and redox-active naphthalene diimide-based (NDI) linkers[5]. Electrochemical studies of Zr(NDI) MOF thin films grown on fluorine-doped tin oxide (FTO) showed that charge propagation through the MOF occurs by a diffusion-like linker-to-linker hopping mechanism, with an apparent diffusion coefficient of $(5.4 \pm 1.1) \times 10^{-11}$ cm$^2$ s$^{-1}$ using KPF$_6$ as the electrolyte. The cyclic voltammograms (CVs) of the Zr(NDI) MOF thin films feature two reversible electrochemical reductions assigned to the NDI$^{0/-}$ and NDI$^{-/2-}$ couples.

In photoelectrochemistry, solar energy is harvested to power redox reactions, capitalizing on absorbed photons to drive those reactions at a lower energetic cost (Fig. 1)[6,7]. The diminished requirement for externally applied voltage is balanced by the photovoltage of the semiconductor, $V_{ph}$, set by the difference between the two quasi-Fermi levels associated with the energy of the holes, $E_{f,p}$, and electrons, $E_{f,n}$, generated under illumination at wavelengths sufficient to excite electrons from the valence to the conduction band. However, the actual photovoltage values often fall short of the maximum obtainable photovoltage due to detrimental recombination events, including surface recombination

processes ($j_{rec}$) that compete with productive interfacial charge transfer ($j_{tr}$)[8–11].

Successful transport of minority charges across the interface is highly sensitive to surface functionalization and attachment strategies[12]. The presence of a redox acceptor in proximity to a photocathode does not guarantee transfer of photogenerated charges, as charge recombination processes must be overcome to translocate the minority carriers across the interface. For example, in a study of the effects of the linker between a cobaloxime catalyst and TiO$_2$-coated Si(100) wafers, no evidence for the reduction of the molecular species in solution at an illuminated photocathode was found, even though the same species was shown to be reduced when immobilized at the photoelectrode[13]. In another example, immobilization of a nickel–phosphine catalyst at a methylated Si(111) surface increased the photovoltage of the functionalized semiconductor by 200 mV compared to the Si photocathode when measured in the presence of the catalyst in solution[14]. In both cases, the increased performance upon immobilization was tentatively assigned to improved interfacial charge transfer to the immobilized species.

Atomic layer deposition (ALD) of transparent oxide coatings such as TiO$_2$ has been widely used to slow charge recombination and protect semiconductors against corrosion in photoelectrochemical applications[15,16]. Formation of a p–n junction at p-GaP using an ALD-deposited layer of n-Nb$_2$O$_5$ and a Pt catalyst resulted in a record photovoltage of 710 mV, an improvement of 360 mV compared to a GaP photocathode with Pt only and no metal oxide layer[9]. Molecular functionalization of semiconductors can also lower barriers for photoelectrochemical reactions[14,17–25]. In the context of metal–organic hybrid constructs, Downes and Marinescu[26] demonstrated photoelectrochemical H$_2$ evolution in H$_2$SO$_4$ solution (pH 1.3) by a cobalt dithiolene polymer composing a metal–organic surface (MOS) dropcast onto p-type Si electrodes. In this study, the electrochemical potential required to achieve equivalent rates of H$_2$ evolution was 550 mV less under illumination than that of the MOS at glassy carbon electrodes. More recently, a Cu-based MOF at a Cu$_2$O photoelectrode operated in acetonitrile was used to enhance the CO$_2$ reduction activity of the Cu$_2$O surface[27].

While these reports show an increase in performance of the MOF-modified semiconductors, it is somewhat unclear whether the improvement is due to the catalytic activity of the surface species themselves or to changes in surface energetics that modulate the band edges and/or improve charge separation[28–30]. In addition, in the case of a 3D material being added to the surface, improvements in catalysis could be due to a higher reaction surface area. To date, clear evidence of photoelectrochemical reduction of a discrete molecular species in a MOF framework has not been reported. Herein, we investigate whether p-type semiconductors can be used to reduce the redox-active linkers that comprise the Zr(NDI) MOF introduced above.

## Results

**Preparation of Zr(NDI) thin films.** ALD-deposited TiO$_2$ has been used as an interface to immobilize molecular catalysts at semiconductors[22–24,31], and in our construct, provides a metal oxide surface for anchoring the carboxylate-functionalized NDI linker. TiO$_2$ was deposited by ALD onto freshly etched semiconductor (SC, where SC is Si or GaP) wafers using 100 alternating TiCl$_4$:H$_2$O cycles at 200 °C, which results in TiO$_2$@SC samples amenable to the attachment of carboxylate groups[32]. The oxide layer was 3.7-nm thick as measured by spectroscopic ellipsometry. The MOF was grown on the TiO$_2$-coated substrate as reported previously for Zr(NDI)@FTO. Briefly, the NDI linkers are first attached to the substrate by exposure of the TiO$_2$-coated

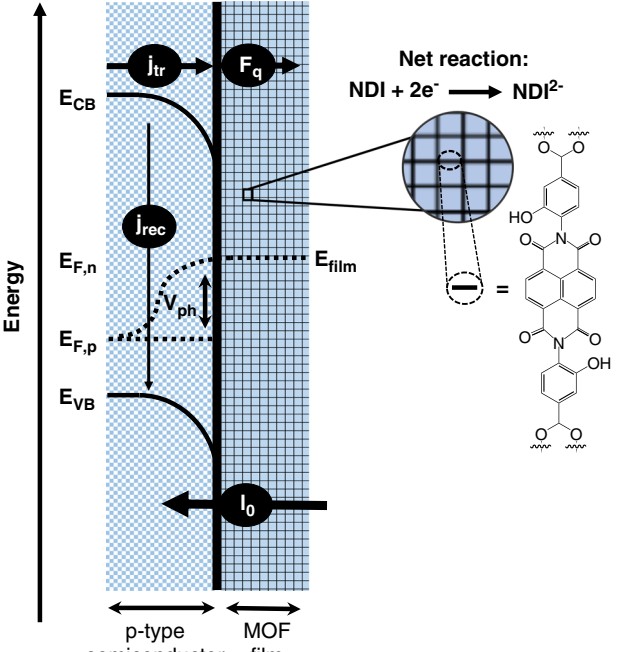

**Fig. 1 Balance of steady-state photon ($I_O$), interfacial electron transfer ($j_{tr}$), recombination ($j_{rec}$), and diffusion-controlled charge ($F_q$) fluxes.** $E_{VB}$ and $E_{CB}$ are the energetic positions of the valence and conduction band, respectively, $E_{F,p}$ and $E_{F,n}$ are the quasi-Fermi levels of the holes and electrons, respectively, $V_{ph}$ is the semiconductor photovoltage, and $E_{film}$ is the electrochemical potential of the MOF film. The molecular structure of the linker is pictured on the right.

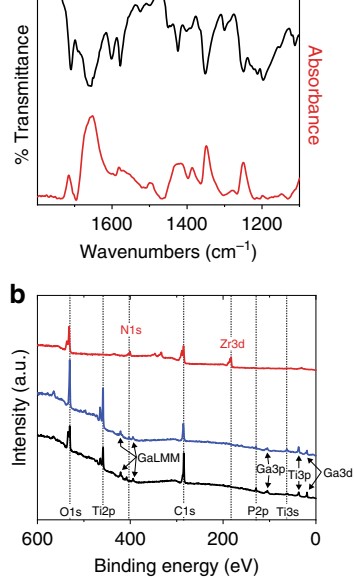

**Fig. 2 Surface characterization of Zr(NDI)|TiO$_2$@GaP. a** ATR-FTIR transmittance spectrum of bulk Zr(NDI) powder (black) and absorbance spectrum of Zr(NDI)|TiO$_2$@GaP (red). **b** Survey XPS spectra of TiO$_2$@GaP (black), NDI|TiO$_2$@GaP (blue), and Zr(NDI)|TiO$_2$@GaP (red). Corresponding characterization for Si substrates can be found in the Supplementary Information.

wafers to a solution of NDI in DMF, resulting in NDI|TiO$_2$@SC. The NDI layer was 2 nm as measured by spectroscopic ellipsometry and using a Cauchy layer model for the NDI film. This is in good agreement with formation of a monolayer of NDI, which has a length of 18 Å. MOF growth on the NDI-coated wafers proceeds in a premixed and sonicated solution of the NDI linker, ZrCl$_4$, and acetic acid in DMF under direct solvothermal synthesis conditions at 120 °C for 3 days, resulting in Zr (NDI)|TiO$_2$@SC.

**Film characterization.** Attenuated total reflection Fourier transform-infrared (ATR-FTIR) spectroscopic studies of the Zr (NDI)|TiO$_2$@SC show that the characteristic peaks of the Zr(NDI) bulk powder sample are retained (Fig. 2a). In particular, and as described previously[5], frequencies associated with the symmetrical and asymmetrical stretches of the imide C = O bond of the NDI linker (≈1650 and 1710 cm$^{-1}$, respectively) appear in the carbonyl region of the spectrum. In addition, peaks indicative of the C–O–C stretch (≈1250 cm$^{-1}$), imide C–N bonds (1350 cm$^{-1}$), and ring deformation of the imide (1420 cm$^{-1}$) are present in the spectra of the SC-grown MOF films as well as those of the bulk MOF and the NDI linker (Supplementary Fig. 1). Functionalization of the SC surface was also examined by X-ray photoelectron spectroscopy (XPS). Spectra of the TiO$_2$-coated SC wafers have peaks corresponding to the binding energies of Ti2p, O1s, and C1s (Fig. 1b and Supplementary Figs. 2 and 3). After exposure to the NDI solution, additional contributions from the nitrogen moieties in the NDI linker are present in the N1s region. Following solvothermal synthesis, the intensity of these contributions increases and new peaks appear in the region of Zr3d binding energy, indicative of the Zr SBUs of the MOF. The ratio of Zr:N is 1:1.8 ± 0.2 (Supplementary Table 1), consistent with each of the hexanuclear Zr nodes sharing twelve linkers in a repeating crystalline fashion. Extensive efforts to characterize the films by X-ray diffraction (XRD) either by grazing-incidence XRD or with long-scan

measurements (70 s/step) were unfortunately unsuccessful, a fact that we ascribe to the thinness of the films and their preferred orientation that precludes observation of XRD. Scanning electron microscopy (SEM), in contrast, provides evidence of the crystallinity of the MOF thin film after directed growth on the titania-coated substrate (Fig. 3 and Supplementary Fig. 5a). Octahedral crystals characteristic of the UiO-type morphology are evident, with a particle size of 100–200 nm. The film thickness is 150 ± 43 nm (where the range arises from sample-to-sample variation) determined by cross sections of the films imaged by SEM (Supplementary Fig. 6). SEM with energy- dispersive X-ray spectroscopy (EDX) data across multiple samples and geometric areas yields a Zr:N ratio of 1:2.0 ± 0.4 (Supplementary Fig. 5b and Supplementary Tables 2 and 3), providing additional evidence for the proposed structure of the film.

**Cyclic voltammetry at illuminated silicon photocathodes.** The photoelectrochemical response of three modified Si electrodes—TiO$_2$-coated Si (TiO$_2$@Si), those with an NDI monolayer grown on the TiO$_2$-coated Si (NDI|TiO$_2$@Si), and the complete Zr(NDI) MOF-functionalized electrode (Zr(NDI)|TiO$_2$@Si)—was tested in a three-electrode configuration using the functionalized Si wafer as the working electrode, a carbon counter electrode, and a Ag wire pseudoreference electrode. In 0.5 M LiClO$_4$ in DMF and under 1-sun illumination, the CV of TiO$_2$@Si has little-to-no photoelectrochemical response at a scan rate of 100 mV s$^{-1}$ (Supplementary Fig. 7). However, under the same conditions, the photocathode functionalized with a monolayer of the NDI linker, NDI|TiO$_2$@Si, exhibits two redox events in the CV (Fig. 4a and Table 1). The midpoint potentials are observed at −1.00 and −1.25 V vs. Fc$^{+/0}$, close to those of the NDI linker on FTO-grown MOF (−0.97 and −1.30 V vs. Fc$^{+/0}$). We thus assign these features to two normally ordered one-electron reductions of the NDI linker to NDI$^-$ and NDI$^{2-}$ (The Zr nodes solely play a coordinative and structural role in the film, without participating in charge transfer itself, as the Zr(IV)/(III) couple is outside the redox window that we are operating within.). After expanding the monolayer into the third dimension in a MOF thin film (Zr (NDI)|TiO$_2$@Si), relatively broad reductive and oxidative peaks are observed at potentials more positive than those in the monolayer-functionalized Si sample ($E_{c,p}$ = −0.99 and $E_{a,p}$ = −0.18 V vs. Fc$^{+/0}$, $v$ = 100 mV s$^{-1}$, Fig. 4a). At slower scan rates ($v$ = 5 mV s$^{-1}$), the two reductions separate into distinct features, with midpoint potentials of −0.64 and −0.78 V vs. Fc$^{+/0}$ (Supplementary Fig. 8). The large peak-to-peak separations (0.43 and 0.46 V) are commonly observed in MOF electrochemistry and are attributed to sluggish interfacial electron transfer, ohmic resistance in the film, and/or slow ion transport at the film|electrolyte interface[33–37].

**Photovoltage of Zr(NDI)|TiO$_2$@Si.** Crucially, the midpoint potentials of the first and second reductions of the NDI linker in Zr(NDI)|TiO$_2$@Si are shifted positive by 330 mV and 520 mV, respectively, compared to the formal potentials of the Zr(NDI) @FTO construct. In photoelectrocatalytic applications, the open-circuit potential ($V_{oc}$) or onset potential ($V_{on}$) is typically reported as a measure of the photovoltage, a value that depends on equilibration between the semiconductor and redox couple, as well as the surface recombination rates and overpotential requirements[9,11,38]. In the case of NDI|TiO$_2$@Si, the lack of a photoeffect on the NDI electrochemical potentials suggests poor charge separation at the semiconductor interface, resulting in the need for a large applied bias to increase band bending in order to populate the conduction band with minority carriers. The similar midpoint potentials observed in the dark electrochemistry of the

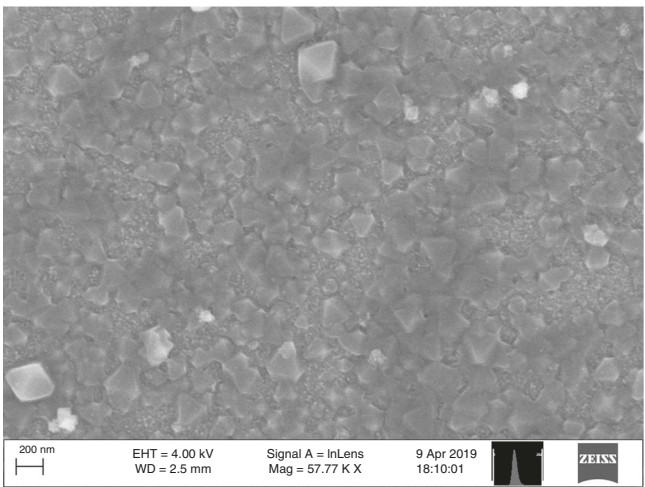

**Fig. 3 Scanning electron microscopy (SEM) of Zr(NDI)|TiO₂@GaP.** SEM image showing octahedra typical of the PIZOF-type MOF. Corresponding characterization for Si substrates can be found in the Supplementary Information.

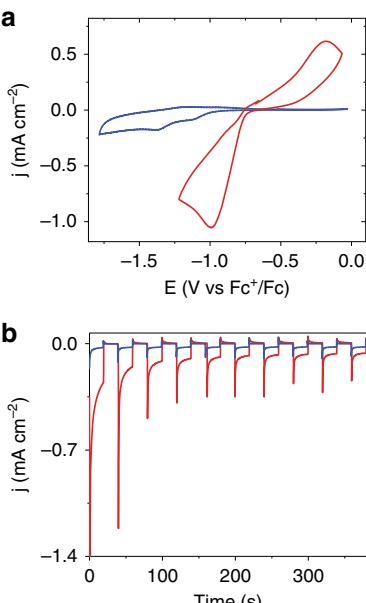

**Fig. 4 Photoelectrochemical characterization of modified Si photocathodes. a** Cyclic voltammograms of NDI|TiO₂@Si (blue) and Zr (NDI)|TiO₂@Si working electrodes (red) at a scan rate of 100 mV s⁻¹ with 0.5 M LiClO₄ in DMF as the supporting electrolyte under AM 1.5 illumination. **b** Chronoamperograms of NDI|TiO₂@Si (blue) and Zr (NDI)|TiO₂@Si (red) working electrodes under chopped light illumination at a potential of −1.1 V vs. Fc⁺/⁰ with 0.5 M LiClO₄ in DMF as the supporting electrolyte in the presence of 1 mM tris(2,2′-bipyridine)cobalt (III) tris(hexafluorophosphate).

film on FTO and illuminated NDI|TiO₂@Si also clarify the role of the TiO₂ layer, which, in our case, serves solely as a tunneling barrier and does not facilitate charge separation or stabilization. In contrast, the decreased potential requirement upon introduction of the MOF layer represents the formation of a hetero-junction capable of providing a photovoltage.

**Electroactive portion of the Zr(NDI) thin film.** The electroactive coverage of NDI linkers was calculated from controlled potential electrolysis experiments by stepping the potential from +0.1 to

−1.2 V vs. Fc⁺/⁰, providing sufficient driving force to fully reduce NDI twice to NDI²⁻ (Supplementary Fig. 10). Analysis of the charge passed using three samples yields an electroactive surface concentration of $68 \pm 42$ nmol cm⁻². For comparison, a tightly packed monolayer of NDI, with the molecule oriented perpendicular to the surface, gives a surface concentration of 1.1 nmol cm⁻². Thus, the extended MOF architecture allows for a significantly higher loading of redox-active sites at the electrode. Nonetheless, simply increasing the number of sites may not always result in a current enhancement, as diffusional processes can become limiting as films become thicker[39]. To calculate the diffusion coefficient of the film, the chronoamperometry data were fitted to the Cottrell equation (Eq. (1))

$$i = \frac{nFSC^0_{\mathrm{NDI}}\sqrt{D_{\mathrm{app}}}}{\sqrt{\pi t}}, \tag{1}$$

where $i$ is current (A), $n$ is the number of electrons, $F$ is Faraday's constant (C mol⁻¹), $S$ is the electrode surface area (cm²), $C^0_{\mathrm{NDI}}$ is the concentration of NDI linkers on the surface (mol cm⁻³), $D_{\mathrm{app}}$ is the apparent diffusion coefficient (cm² s⁻¹), and $t$ is time (s). This analysis results in a diffusion coefficient of $(1.1 \pm 1.2) \times 10^{-10}$ cm² s⁻¹. For comparison, $D_{\mathrm{app}}$ for the NDI film at FTO was found to be $(5.4 \pm 1.1) \times 10^{-11}$ cm² s⁻¹ using KPF₆ as the electrolyte, and more recent investigations have found that the diffusion coefficient for these constructs can be increased by one order of magnitude using LiClO₄ as the electrolyte (Supplementary Fig. 11). Thus, the apparent diffusion coefficient for the Zr (NDI) MOF film (measured with LiClO₄) is in the same order of magnitude in dark electrochemistry experiments at conducting substrates as on illuminated semiconductors. This provides further evidence for the successful formation of the proposed Zr (NDI) MOF (i.e., not only the molecular identity but also the macroscopic properties of the film are conserved) and indicates that charge transport properties through the film are unchanged, regardless of the substrate or film thickness. The similar apparent diffusion coefficients also suggest that the photocurrents in Zr (NDI)|TiO₂@Si are largely determined by the charge transfer characteristics of the MOF.

The amount of the film that is electrochemically probed can be calculated using Eq. (2)[40–42]

$$\delta = \sqrt{\frac{D_{\mathrm{app}}\mathrm{RT}}{Fv}}, \tag{2}$$

where $\delta$ is the diffusion-layer thickness (cm), $D_{\mathrm{app}}$ is the apparent diffusion coefficient (cm² s⁻¹), $R$ is the gas constant (J mol⁻¹ K⁻¹), $T$ is temperature (K), $F$ is Faraday's constant (C mol⁻¹), and $v$ is the scan rate (V s⁻¹). With an apparent diffusion coefficient of $10^{-10}$ cm² s⁻¹ and a scan rate of 100 mV s⁻¹, about 50 nm of the film is probed during the timescale of the measurement. At slower scan rates, all of the film can be electrochemically accessed as the diffusion-layer thickness increases, approaching the MOF film boundary to reach a regime of finite diffusion[41]. In this regime, the current is given by Eq. (3)

$$i_{\mathrm{pc}} = FSC^0_{\mathrm{NDI}}d_f\frac{Fv}{4\mathrm{RT}}, \tag{3}$$

where $i_{\mathrm{pc}}$ is the cathodic peak current (A), $F$ is Faraday's constant (C mol⁻¹), $S$ is the electrode surface area (cm²), $C^0_{\mathrm{NDI}}$ is the concentration of NDI linkers on the surface (mol cm⁻³), $d_f$ is the film thickness, $v$ is the scan rate (V s⁻¹), $R$ is the gas constant (J mol⁻¹ K⁻¹), and $T$ is temperature (K). At a scan rate of 5 mV s⁻¹, the diffusion-layer thickness is >200 nm (Eq. (2)), larger than the average film thickness (see above and Supplementary Fig. 6). Under these conditions, the observed current magnitude of −0.19 mA cm⁻² (Supplementary Fig. 8b) is within the theoretical

range of $-0.29 \pm 0.14$ mA cm$^{-2}$, calculated with a finite-diffusion model (Eq. (3)).

Applications of MOF coatings at semiconductors for reductive fuel-forming reactions depend on the ability of the film to not only capture minority carriers but also to transfer those charges to the substrate. Thus, charge transfer through the film was further investigated in the presence of an electron acceptor in the electrolyte solution that is thermodynamically capable of being reduced by electrons from the MOF film. The addition of tris (2,2′-bipyridine)cobalt(III) tris(hexafluorophosphate) to the electrolyte solution (1 mM in DMF with 0.5 M LiClO$_4$) results in increased photocurrents under chopped light chronoamperometry conditions (Supplementary Fig. 12a) compared to experiments with no external electron acceptor (The measured photocurrents are due to the underlying SC in the Zr(NDI)|TiO$_2$@SC films, as reference experiments on Zr(NDI)@TiO$_2$ show photocurrents <0.5 uA/cm$^2$ (see Figure S16 for comparison).). This illustrates that photogenerated charges diffuse through the entire redox-active film and can be extracted at the film|solution interface. Stirring the solution does not further increase the photocurrent (Supplementary Fig. 12b), indicating that the photocurrent is not limited by a diffusional concentration gradient of the Co(III) acceptor at the film|solution interface. The importance of the Zr (NDI) MOF in the experiments is evident by a >300% increase in photocurrent of the MOF-modified samples compared to those of NDI-only modified photocathodes (Fig. 4b). Spikes in the chronoamperogram upon illumination are characteristic of charge transfer to surface states, followed by a decay to steady-state current that reflects the ratio of interfacial charge transfer ($j_{tr}$) to recombination ($j_{rec}$)[43]. The appearance of transient photocurrents evident under chopped light illumination suggests that the photovoltage could be further increased through engineering of future MOFs that focus on slowing recombination processes at the interface.

**Photovoltage of Zr(NDI)|TiO$_2$@GaP.** To demonstrate the broader applicability of MOF thin films as surface coatings to increase photovoltage, we extended this work to p-type GaP, a III–V semiconductor with conduction and valence band positions that fulfill the energetic requirements for both hydrogen and oxygen evolution. The photoelectrochemical responses of TiO$_2$@GaP, NDI|TiO$_2$@GaP, and Zr(NDI)|TiO$_2$@GaP were evaluated. In a 0.5 M LiClO$_4$ DMF solution under 1-sun illumination, TiO$_2$@GaP and NDI|TiO$_2$@GaP samples show similar PEC responses (Supplementary Fig. 13). In the dark, little-to-no current passes, while upon illumination, a slight increase in current is observed at negative potentials ($-0.8$ V vs. Fc$^{+/0}$). In contrast, the MOF-modified samples show two clearly defined redox waves upon illumination, with midpoint potentials at remarkably positive values of $-0.26$ and $-0.50$ V vs. Fc$^{+/0}$ (Fig. 5 and Table 1). The first and second reductions are shifted anodically by 710 mV and 800 mV, respectively, compared to the formal potentials of Zr(NDI)@FTO (Supplementary Figs. 14 and 15). These results illustrate that the Zr(NDI) film effectively captures photogenerated charges at the MOF|GaP interface, with the semiconductor playing an integral role in harnessing solar energy to drive the reduction of the NDI linkers. In addition, the two one-electron reductions confirm that the molecular identity of the MOF-immobilized NDI linkers is preserved. Finally, the Zr (NDI)|TiO$_2$ coating also provides ample protection against hydrolytic degradation, a crucial consideration in solar fuel applications. When tested in aqueous conditions (pH 8.5), the NDI-based reductions are still visible with similar photovoltage, but now as one wave due to potential inversion of the two

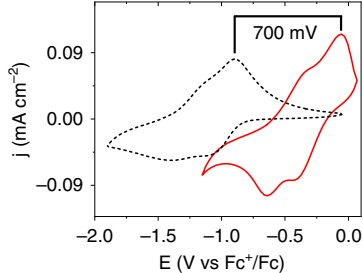

**Fig. 5 Photoelectrochemical characterization of modified GaP photocathodes.** Cyclic voltammograms of Zr(NDI)@FTO (black dashed) and an illuminated Zr(NDI)|TiO$_2$@GaP working electrode (red) at a scan rate of 100 mV s$^{-1}$ with 0.5 M LiClO$_4$ in DMF as the supporting electrolyte. The red solid data were collected under AM 1.5 illumination.

**Table 1 (Photo)electrochemical data (volts vs. Fc$^{+/0}$).**

|  | $E_{1/2}{}^{0/-}$ ($\Delta E_p$ [mV]) | $E_{1/2}{}^{-/2-}$ ($\Delta E_p$ [mV]) |
|---|---|---|
| NDI | $-0.93$ (71) | $-1.21$ (74) |
| Zr(NDI) |@FTO | $-0.97$ (144) | $-1.30$ (190) |
| NDI |TiO2@Si* | $-1.00$ (225) | $-1.25$ (208) |
| Zr(NDI) |TiO2@Si* | $-0.64$ (434) | $-0.78$ (460) |
| Zr(NDI) |TiO2@GaP* | $-0.26$ (380) | $-0.50$ (310) |

*collected under 1-sun illumination.

processes as observed earlier in Zr(NDI)@FTO in aqueous electrolyte (Supplementary Fig. 17)[5].

Photovoltages of up to 710 mV have previously been reported for GaP in photoelectrochemical applications, and have been shown to depend on a variety of factors, including the nature of back contacts, formation of a p–n junction, surface-preparation techniques, and/or surface modification[9,44]. In our construct, the reduction of NDI-based linkers within a surface-immobilized MOF occurs at potentials over 700 mV positive of the formal potentials at a conductive substrate. This state-of-the-art photovoltage is achieved by employing a redox-active moiety within a MOF thin film, enabling charges at the interface to be efficiently captured and carried away by molecular components, thereby lessening the probability of charge recombination. An extended spatially ordered network of redox-active linkers aids charge separation through localization of charges away from the interface where detrimental charge recombination processes can occur. This results in the highest photovoltage reported for a GaP photocathode modified with only earth-abundant materials.

**Effect of light intensity on photocurrent of Zr(NDI)|TiO$_2$@SC.** The current response at a film-modified photoelectrode may be limited by the number of photons converted to interfacial charge carriers (the light absorption, charge separation, and charge transfer processes)[45] or by the diffusion of those charge carriers to redox centers. At both semiconductors (SCs, p-type Si, and GaP), a threshold of light intensity must be reached to produce photocurrent, indicating that light is required to reduce Zr(NDI)|-TiO$_2$@SC. Remarkably, decreasing the light intensity from 100 mW cm$^{-2}$ (1 sun) to 60 or 10 mW cm$^{-2}$ (for GaP or Si substrates, respectively,) does not decrease the photocurrent of the photoelectrodes (Supplementary Figs. 18 and 19). The lack of linear dependence of the peak current to light intensity indicates that the rate of charge propagation through the film is not controlled by photon flux, but by diffusional electron transport through the MOF, as described above.

## Discussion

In this report, we have shown that not only are photogenerated charges effectively transported from a visible-light-absorbing semiconductor to the first layer of molecular linkers in a MOF film, but that the extended framework allows those charges to be extracted at a lower applied potential compared to a conductive electrode. We provide unequivocal electrochemical evidence of the interfacial charge transfer of photogenerated electrons from visible-light-absorbing semiconductors to a MOF-confined molecular species, and highlight the role of the MOF in promoting charge separation and improving the photovoltage of the semiconductor–MOF hybrid material. The improved photovoltage caused by the MOF coating is likely because of the capture and spatial separation of the minority carriers across the film. We show that the charges captured by the film can be translocated through the MOF by a hopping mechanism between discrete molecular moieties, and finally extracted by an electron acceptor in solution. This highlights the promise of this construct to not only accumulate charges at a mild potential, but also to serve as an electron reservoir for further reductive reactions. The lack of photocurrent dependence on light intensity, along with the lack of current increase upon stirring the solution with an external electron acceptor, indicates that the current is not limited by photon flux or mass transport at the film|solution interface. The calculated diffusion coefficient for the film evaluated at an illuminated semiconductor substrate is similar to that of the film grown at a conductive substrate[5], suggesting that the current is limited by the same process, i.e., diffusion of charge through the film. Additional evidence that the photocurrent is under diffusional control is found through modeling the current under finite-diffusion conditions.

P-type Si and GaP have previously been used as substrates for hydrogen evolution and carbon dioxide reduction[20–23,44,45]. Our findings indicate that the advantages of MOFs, such as their high surface area and porosity combined with their ability to house and stabilize molecular catalysts, may be integrated with SCs for MOF-based photoelectrosynthetic applications. Ion-permeable coatings allow key characteristics such as photovoltage to be optimized by forming adaptive junctions at the interface[38]. We have shown that the construct can be operated in aqueous conditions, a promising feature in the context of light-driven hydrogen evolution. In particular, MOF thin films can serve as novel semiconductor coatings to enhance photovoltages by promoting interfacial charge separation of photogenerated charges.

## Methods

**General details**. All reagents were purchased from commercial sources and used as received, except for DMF, which was stored over activated 3Å molecular sieves. B-doped Si (111) substrates (ITME) had a thickness of 265 uM and a resistivity of 0.3–0.5 Ω cm. Zn-doped GaP(100) substrates (ITME) had a thickness of 420 μM, with a resistivity of 0.15–0.16 Ω cm, a dopant concentration of $5.0–5.5 \times 10^{17}$ cm$^{-3}$, and an etch-pit density of $<5 \times 10^{4}$ cm$^{-2}$. Before ALD deposition, Si wafers were cleaned by three successive 5-min sonications in acetone, dichloromethane, and water. The wafers were then etched in ammonium fluoride for 20 min, followed by rinsing with water and storing in an argon environment. GaP wafers were cleaned by sonication for 5 min in acetone, followed by 5 min of sonication in isopropanol. The wafers were then etched in concentrated $H_2SO_4$ for 30 s and rinsed with ethanol. $TiO_2$ was deposited at 200 °C using a Picosun R-200 ALD system. The film was grown with alternating cycles of the $TiCl_4$ and $H_2O$ precursors. The synthesis of the linker NDI and solvothermal synthesis of Zr(NDI) have been previously described[5]. After solvothermal synthesis, the SC wafers were sonicated for 10 min in DMF.

**Instrumental details**. ATR-FTIR data were collected on a Bruker ALPHA FTIR spectrometer from 4000 to 450 cm$^{-1}$ at room temperature. XPS was performed on a PHI Quantera SXM using a monochromatic Al Kα source (hν = 1486.6 eV) operated at 25 W with a beam diameter of 100 μM and a pass energy of 224 eV. Data were analyzed using Multipak or Casa software, and all spectra calibrated by setting adventitious carbon to 284.8 eV. SEM was performed with a Zeiss 1550 SEM with InLens detector. The measurements were performed at 4–5-kV accelerating voltage at working distances of 3–5 mm. ImageJ (NIH) software was used to measure the film thickness. EDX data were collected with an 80-mm$^2$ Silicon Drift Detector using AZtec (INCA energy) software at an acceleration voltage of 5 kV and a working distance of 6.5 nm or 8.5 mm.

**Photoelectrochemical measurements**. Photoelectrochemical measurements were performed in a three-electrode configuration using a custom cell (Pine Instruments) equipped with a quartz window. Working electrodes were prepared by scratching the back of the wafer with GaIn eutectic (Sigma-Aldrich). Electrical contact to the stainless-steel back contact was made with a piece of conductive copper tape (Electron Microscopy Sciences). The counter electrode was a glassy carbon rod, and a Ag wire was used as a pseudoreference that was calibrated against Fc$^{+/0}$ at the beginning and end of every experimental data set. The illumination source was an Enli Tech Solar Simulator with Xe light source and an AM 1.5 G spectral correction filter. The light intensity was calibrated using a NREL-traceable silicon reference cell.

## Data availability

The data that support the findings of this study are available from the corresponding author upon reasonable request.

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

## Acknowledgements

Financial support from European Research Council (ERC-CoG2015-681895_MOFcat) and the Swedish Energy Agency is gratefully acknowledged. A.M.B was supported by the Olle Engkvist Byggmästare Foundation.

## Author contributions

A.M.B. synthesized the construct, acquired, analyzed, interpreted the data, and drafted the work. B.D.M. assisted in data collection, analysis, and interpretation. B.A.J. contributed to the electrochemical experiments and interpretation of the work. S.O. contributed to the conception, design, and interpretation of the work. All authors contributed to the interpretation of data and the writing of the paper. All authors have given approval to the final version of the paper.

## Funding

## Competing interests

The authors declare no competing interests.
