## [Peer Review File · Nature Communications]

Reviewers' comments:

Reviewer #1 (Remarks to the Author):

The manuscript presents the deposition of a previously studied NDI-based MOF (J. Am. Chem. Soc.) on the surface of GaP(100) for use as a modified photocathode.

To address literature precedence - The deposition of MOFs on visible light semiconductors has certainly been explored in the literature. Additionally, the photoelectrochemical response of MOFs has been studied. The difference in the submitted work is that the photoelectrochemical action comes from the underlying GaP semiconductor and the MOF acts as an electron-accepter. This phenomenon has been explored a copper-based MOF on copper oxide for carbon dioxide reduction (J. Am. Chem. Soc. 2019).

To address the scientific results and presentation of those results -

MOF characterizations - The PXRD leaves much to be desired. Have the authors calculated the oriented PXRD patterns and compared to see what preferential orientation is observed? The intensity of the peaks also differs greatly from the predicted pattern. Are there peaks from the underlying GaP? The SEM provides evidence of MOF crystallites. However, they do not appear to be oriented, but indeed polycrystalline.

Electrochemical characterizations -

[Formatting] It is not clear in the Tables and Figures that the data is collected under illumination. Additionally, some figures are presented in a dashed format, but the figure legend does not refer to the line style.

[Science] The comparison of the illuminated photocathode to dissolved species seems odd. Why not compare to the MOF on a non-light activated substrate? The authors certainly have this data from the previous J. Am. Chem. Soc. paper and it would be a more appropriate comparison. Is the shift as pronounced in this comparison? The reviewer would guess, yes.

Discussion - The photovoltage should be set by the difference between the redox potential in the dark and the Fermi level of the GaP (as indicated in Scheme 1). These parameters should be quantifiable in control experiments (CV in dark at conductive electrode, XPS). Do the numbers observed agree with the observed photovoltage? Also, it would be of great benefit to add numbers to Scheme 1 to support the observed effect.

Overall, the results are in line with what would be expected - i.e. the reviewer is not surprised by the science. There are only minor scientific components that should be addressed to render the manuscript publishable. Given the literature precedence and the routine nature of the observations, the reviewer does not feel the manuscript rises to the level of a Nature Communications article. The manuscript is more suited for Chemical Communications or similar.

Reviewer #2 (Remarks to the Author):

In this work, the authors show that a redox-active MOF can be photo-electrochemically reduced when immobilized at a semiconductor GaP surface. They show that the reduction of Redox-active linkers in a GaP-immobilized MOF occurs at potentials ≈ 700 mV positive of the reduction potentials at a non photo-active substrate, due to effective capture of interfacial electrons and subsequent GaP-generated photovoltage. These findings are indeed very interesting and could be of importance for future development of MOF-based solar fuel generation schemes. However,

the presented results are too preliminary for publication in Nature Communications at this point, and considerable amount of further experimental work is still needed.

1) The authors claim that the 700 mV shift in MOF redox activity is a result of GaP photovoltage, yet no measurements of open-circuit potential under illumination is provided to prove this notion.

2) In Figure S4, the authors compare between the redox activity of GaP and FTO immobilized MOF, showing the resulting 700 mV shift. however, the currents are normalized and it is not possible to understand if the recorded currents were similar or much different. In other words, it is already known that with FTO, almost all MOF-linkers are electroactive, but no data is provided to let us understand what fraction of MOF-linkers are electroactive at Gap under illumination. It could be that only a single layer of linkers close to the GaP are active and thus the utilization of the MOF is not beneficial in this case. The authors should elaborate on this issue.

3) In Figure S5, the authors compare the CVs under different light illumination intensities, showing that the current is not dependent on illumination intensity. Yet, In lower illumination intensities (e.g. 0.1 Sun), a decrease in photovoltage is expected and thus the potential shift of redox peaks showed be lower accordingly. The authors should provide more insights on this issue.

4) Are redox hopping rates similar with FTO and Gap substrates? A comparison should be included.

Reviewer #3 (Remarks to the Author):

In this manuscript, the authors fabricated MOF/TiO₂/GaP electrodes and investigate the photoelectrochemical reduction on them. However, it is very difficult to find scientific advantage of the MOF on the semiconductors for the photoelectrochemical reactions. I recommend submitting to more specialized journals. Before the submission, more characterization is necessary such as impedance measurements to characterize the heterojunction and thickness dependence of MOF, etc.

We want to stress that the present manuscript has rather little resemblance to the version submitted under NCOMMS-19-25827A-Z. Nevertheless, we try to respond to the original reviewers' comments hereafter in the usual point-by-point fashion.

Reviewer #1:

The manuscript presents the deposition of a previously studied NDI-based MOF (J. Am. Chem. Soc.) on the surface of GaP(100) for use as a modified photocathode.

To address literature precedence - The deposition of MOFs on visible light semiconductors has certainly been explored in the literature. Additionally, the photoelectrochemical response of MOFs has been studied. The difference in the submitted work is that the photoelectrochemical action comes from the underlying GaP semiconductor and the MOF acts as an electron-accepter. This phenomenon has been explored a copper-based MOF on copper oxide for carbon dioxide reduction (J. Am. Chem. Soc. 2019).

We don't dispute that MOFs on semiconductors have been explored in the literature. What has to the best of our knowledge not been demonstrated is direct evidence for the reduction of molecularly defined linkers in the MOF under visible light illumination, for example, by cyclic voltammetry.

The paper that the reviewer refers to (JACS 2019, 10924) reports a current increase for the photoelectrochemical reduction of CO₂ at a constant applied potential. This could indeed be due to the presence of a photovoltage, but frankly also due to other phenomena such as surface alterations for the underlying copper oxide catalyst, increased surface area effects, etc...

In fact, the results that we present in our manuscript support the hypothesis of a photovoltage being present also in the Cu₂O-MOF system in JACS 2019, 10924.

To address the scientific results and presentation of those results -

MOF characterizations - The PXRD leaves much to be desired. Have the authors calculated the oriented PXRD patterns and compared to see what preferential orientation is observed? The intensity of the peaks also differs greatly from the predicted pattern. Are there peaks from the underlying GaP? The SEM provides evidence of MOF crystallites. However, they do not appear to be oriented, but indeed polycrystalline.

Obtaining conclusive PXRD data has been a major challenge in this work. Despite numerous attempts, including efforts by TEM, no good diffraction pattern could be obtained. We attribute this shortcoming to the small amount of material that is present in the ca. 150 nm thick films. To compensate for the lack of diffraction data, we have characterized our films by ATR-FTIR, XPS, SEM-EDX, and top-view and cross section SEMs. In particular, we have examined more closely the elemental ratios by XPS and SEM-EDX (now reported in the manuscript). Both methods provide consistent evidence for the proposed MOF. Also, all electrochemical studies are consistent with the film being the PIZOF-type MOF (as reported in JACS 2018, 2985)

Electrochemical characterizations -

[Formatting] It is not clear in the Tables and Figures that the data is collected under illumination. Additionally, some figures are presented in a dashed format, but the figure

legend does not refer to the line style.

Most Figures and Tables have changed entirely, and extra care was taken with the figure legends.

[Science] The comparison of the illuminated photocathode to dissolved species seems odd. Why not compare to the MOF on a non-light activated substrate? The authors certainly have this data from the previous J. Am. Chem. Soc. paper and it would be a more appropriate comparison. Is the shift as pronounced in this comparison? The reviewer would guess, yes.

We agree with this comment, and all photoelectrochemical data have been compared to dark electrochemical responses on the FTO-grown films.

Discussion - The photovoltage should be set by the difference between the redox potential in the dark and the Fermi level of the GaP (as indicated in Scheme 1). These parameters should be quantifiable in control experiments (CV in dark at conductive electrode, XPS). Do the numbers observed agree with the observed photovoltage? Also, it would be of great benefit to add numbers to Scheme 1 to support the observed effect.

We note that the method of determining photovoltage by comparing the redox potentials at an illuminated semiconductor compared to those in dark electrochemistry has a long scientific tradition (Wrighton et al, JACS, 1980, 102, 3683), and is the most direct way of illustrating the photovoltage. We have expanded the introduction to include a section on the photovoltage, including descriptions of theoretical maximum photovoltages as well as a discussion of why the actual photovoltage usually falls short.

Overall, the results are in line with what would be expected - i.e. the reviewer is not surprised by the science. There are only minor scientific components that should be addressed to render the manuscript publishable. Given the literature precedence and the routine nature of the observations, the reviewer does not feel the manuscript rises to the level of a Nature Communications article. The manuscript is more suited for Chemical Communications or similar.

We admit that the content of the original submission was somewhat “slim”, and have thus greatly expanded the manuscript in scope and altered in focus. The additional findings that have been included in the present version are detailed in the accompanying letter above.

Reviewer #2:

In this work, the authors show that a redox-active MOF can be photo-electrochemically reduced when immobilized at a semiconductor GaP surface. They show that the reduction of Redox-active linkers in a GaP-immobilized MOF occurs at potentials ≈ 700 mV positive of the reduction potentials at a non photo-active substrate, due to effective capture of interfacial electrons and subsequent GaP-generated photovoltage. These findings are indeed very interesting and could be of importance for future development of MOF-based solar fuel generation schemes. However, the presented results are too preliminary for publication in Nature Communications at this point, and considerable amount of further experimental work is still needed.

We admit that the content of the original submission was somewhat “slim”, and have thus greatly expanded the manuscript in scope and altered in focus. The additional findings that have been included in the present version are detailed in the accompanying letter above.

1) The authors claim that the 700 mV shift in MOF redox activity is a result of GaP photovoltage, yet no measurements of open-circuit potential under illumination is provided to prove this notion.

We note that the method of determining photovoltage by comparing the redox potentials at an illuminated semiconductor compared to those in dark electrochemistry has a long scientific tradition (Wrighton et al, JACS, 1980, 102, 3683), and is the most direct way of illustrating the photovoltage. We have expanded the introduction to include a section on the photovoltage, including descriptions of theoretical maximum photovoltages as well as a discussion of why the actual photovoltage usually falls short. We have also added a description of other ways in the photoelectrochemistry literature used to calculate photovoltage (V_{on} and V_{oc}).

2) In Figure S4, the authors compare between the redox activity of GaP and FTO immobilized MOF, showing the resulting 700 mV shift. however, the currents are normalized and it is not possible to understand if the recorded currents were similar or much different. In other words, it is already known that with FTO, almost all MOF-linkers are electroactive, but no data is provided to let us understand what fraction of MOF-linkers are electroactive at Gap under illumination. It could be that only a single layer of linkers close to the GaP are active and thus the utilization of the MOF is not beneficial in this case. The authors should elaborate on this issue.

We have investigated this point in great detail for the Zr(NDI) MOF on Si, including numerical calculations . The findings are presented on page 6 to 10.

3) In Figure S5, the authors compare the CVs under different light illumination intensities, showing that the current is not dependent on illumination intensity. Yet, In lower illumination intensities (e.g. 0.1 Sun), a decrease in photovoltage is expected and thus the potential shift of redox peaks showed be lower accordingly. The authors should provide more insights on this issue.

We have done substantial studies, including numerical simulations of the current response under illumination, and conclude that the current response is mainly determined by the electron transport characteristics of the MOF thin film, thereby explaining the lack of light dependency.

4) Are redox hopping rates similar with FTO and Gap substrates? A comparison should be included.

This is a very good point! The apparent diffusion coefficient for the MOF thin film on illuminated p-Si is indeed very similar to that found for the FTO construct. This finding has been added on page 8.

Reviewer #3:

In this manuscript, the authors fabricated MOF/TiO₂/GaP electrodes and investigate the photoelectrochemical reduction on them. However, it is very difficult to find scientific advantage of the MOF on the semiconductors for the photoelectrochemical reactions. I recommend submitting to more specialized journals. Before the submission, more characterization is necessary such as impedance measurements to characterize the heterojunction and thickness dependence of MOF, etc.

We admit that the content of the original submission was somewhat “slim”, and have thus greatly expanded the manuscript in scope and altered in focus. The additional findings that have been included in the present version are detailed in the accompanying letter above.

Reviewers' comments:

Reviewer #1 (Remarks to the Author):

The authors adequately addressed my scientific concerns. The authors acknowledged in their initial submission the results were "slim". The reviewer does not feel that the current additions elevate the manuscript to the level of Nature Communications.

Over the review rounds, each reviewer commented on the literature precedence and lack of impact of the presented work. The improvements made by the authors to date do not serve to elevate the article to the level of Nature Communications.

Reviewer #2 (Remarks to the Author):

The authors have now expanded their manuscript, giving essential characterizations and scientific discussions which were missing from the previous version.

They have carefully addressed all comments and questions raised by me as well as the other reviewers.

I believe that the manuscript delivers an important notion regarding the factors determining the operation of MOF-semiconductor based photo-electrochemical cells. Thus, I now recommend publication in Nature Communications.

REVIEWER COMMENTS

Reviewer #4 (Remarks to the Author):

This manuscript describes the use of MOFs as coating of to improve the photovoltage of some p-type semiconductors. I recommend the acceptance of this article after major revisions. I provide to the authors some comments and remark some that should be taking into account for its further publication.

In the photoelectrochemistry characterization the Zr(NDI)/TiO₂/FTO electrode must be included for comparison.

Optical properties of the ligand and MOF must be also performed in order to understand the light induced mechanism. In addition, is important to know which are the band diagram of the three components to a better understanding of the charge transfer processes.

X-Ray diffractogram of the film must be include to assure the purity of the MOF and discard the appearance of other phases.

In figure 1b, which are the assignment of the peaks near to N1s in black and blue spectra?

Figure S2 must be change for an improved picture and also to increase the size.

In figure S3, which are de assignments of the shoulder in C1s peak in MOF Zr(NDI) sample and most important which is the assignment of the shoulder in Zr3d signal for the same sample (black dashed). In addition, the high resolution XPS spectra for the electrodes with Si must be also included.

Moreover, the ration between Zr:N ratios; Zr:Ti and Zr:Si or Zr:Ga measured by XPS must be also included to determine the change in the photocurrent (between all samples) whit the amount of MOF. Valence band (from XPS or UPS) must be also included because will give important information about the electronic structure of these electrodes.

In figure S5 the scale bar is not clear and the calculated tetchiness seem more homogenous, than the reported range (100-200 nm).

From your experiments seems that NZr(NDI) MOF is an electroactive material that present a better performance than when you use the bare ligand, even when the redox mechanism is the same in both cases. Is this true? In this case, which is the Zr contribution? Participate in the charge transfer processes? Is Zr playing a role in redox processes with electron acceptors?

The characterization of the films after reaction (XPS and XRD) must be included to evaluate the stability of this systems. In the case of XPS, is necessary to include the high resolution spectra.

Is clear that light play a role in the process, but is very surprising that the redox events are not controlled by the flux. More experiments must be performed (even to lower light intensity) a and without light in order to assure this behavior and to understand better the light induced charge processes in you system.

Referee 4:

This manuscript describes the use of MOFs as coating of to improve the photovoltage of some p-type semiconductors. I recommend the acceptance of this article after major revisions. I provide to the authors some comments and remark some that should be taking into account for its further publication.

In the photoelectrochemistry characterization the Zr(NDI)/TiO₂/FTO electrode must be included for comparison.

We thank the reviewer for this comment, and have included an additional supplemental figure (included here and as Figure S16) monitoring the current of a Zr(NDI)@FTO electrode under chopped light illumination. The experiment clearly shows that the photoelectrochemical response of the Zr(NDI) /TiO₂@SC is due to the underlying semiconductor, as photocurrents in the Zr(NDI)@FTO is <0.5 $\mu\text{A}/\text{cm}^2$ compared to those of Zr(NDI) /TiO₂@SC which are >100 $\mu\text{A}/\text{cm}^2$.

Figure S16. Chronoamperogram of Zr(NDI)@FTO at a bias potential of -1.2 V vs $\text{Fc}^{+/0}$ with 0.5 M LiClO_4 in DMF as the supporting electrolyte under chopped light illumination (100 mW cm^{-2} , shaded areas). The difference in current between dark and light conditions is $<0.5 \mu\text{A cm}^{-2}$, illustrating that the photocurrent response for the MOF immobilized at Si or GaP (Figure S12) is dominated by the underpinning semiconductor.

Footnote 44 has been added to the manuscript to point out this fact:

The measured photocurrents are due to the underlying SC in the Zr(NDI) /TiO₂@SC films as reference experiments on Zr(NDI)@TiO₂ show photocurrents $<0.5 \mu\text{A}/\text{cm}^2$ (see Figure S16 for comparison).

Optical properties of the ligand and MOF must be also performed in order to understand the light induced mechanism. In addition, is important to know which are the band diagram of the three components to a better understanding of the charge transfer processes.

The optical properties of the ligand and of the MOF film immobilized at a transparent electrode have been reported previously in Johnson, et al., J. Am. Chem. Soc. 140, 2985–2994 (2018). We have included a band diagram of the key components in the SI (Figure S19).

X-Ray diffractogram of the film must be included to assure the purity of the MOF and discard the appearance of other phases.

For over a year we attempted to obtain PXRD data for these films. These attempts included long-scan measurements (more than 70 seconds/step) as well as grazing incidence XRD. Unfortunately, in no cases did we obtain satisfactory results. In contrast, the higher resolution SEM images again and again (Figure 1C from manuscript below as an example) showed the classic octahedral UiO-type morphology as expected for this NDI MOF (based on our prior work, see Figure 2B of JACS 2018, 140, 2985):

Also, all other characterization data (XPS, ATR-FTIR, cyclic voltammetry) support successful formation of the MOF.

We ascribe the inability to obtain unambiguous XRD data mainly to two factors; the low film thickness of around 150 nm and a preferred orientation of the MOF thin film along an axis that gives weak reflections. These explanations are supported by currently unpublished data from the group where thicker films could be grown by an alternative method. Films prepared under these conditions are $> 1 \mu\text{m}$ thick (as determined by SEM), and start giving XRD patterns. We include the unpublished XRD herein for the reviewer:

*The XRD patterns of these relatively thick films show a high preferred orientation that disfavors diffraction of the (111) peak, which is the most intense peak in the simulated diffraction pattern. Thus, we conclude that, given such low intensity peaks for a 1 μM film, we are unable to detect diffraction patterns for the 100-200 nm films reported in this work. Based on this data, we can conclude that the MOF thin films used in this manuscript are simply too thin to see diffraction by XRD. This has precedent in the MOF thin film community (see, for example Deng, X. et al, *J. Am. Chem. Soc.* 141, 10924–10929 (2019)).*

To clarify this to the reader, we have added the following text to the manuscript:

Extensive efforts to characterize the thin films by X-ray diffraction (XRD) either by grazing incidence XRD or with long-scan measurements (70 seconds/step) were unfortunately unsuccessful, a fact that we ascribe to the thinness of the films and their preferred orientation that precludes observation of X-ray diffraction.

In figure 1b, which are the assignment of the peaks near to N1s in black and blue spectra?

These peaks arise from the Ga LMM lines. They have been labeled within Figure 1b for clarification.

Figure S2 must be change for an improved picture and also to increase the size.

Figure S2 has been enlarged and the image quality improved.

In figure S3, which are de assignments of the shoulder in C1s peak in MOF Zr(NDI) sample and most important which is the assignment of the shoulder in Zr3d signal for the same sample (black dashed). In addition, the high resolution XPS spectra for the electrodes with Si must be also included.

The shoulder at 288.3 eV arises from carbons in the imide environment. The shoulder in the Zr 3d spectrum is likely due to sub-oxides in the bulk MOF sample. Notably, these are not apparent in the MOF films grown on the semiconductors. High-resolution spectra of Si electrodes have also been included, and all figures have been remade with additional labels for clarity.

Moreover, the ration between Zr:N ratios; Zr:Ti and Zr:Si or Zr:Ga measured by XPS must be also included to determine the change in the photocurrent (between all samples) whit the amount of MOF. Valence band (from XPS or UPS) must be also included because will give important information about the electronic structure of these electrodes.

The Zr:N ratios are reported in the manuscript ($1:1.8 \pm 0.2$) and we have added a table with the data points in the SI (Table S1). As XPS is a surface-sensitive technique, we are unable to calculate the Zr ratios to any of the substrate elements, or to obtain valence-band region spectra that are representative of the underlying semiconductor.

In figure S5 the scale bar is not clear and the calculated tetchiness seem more homogenous, than the reported range (100-200 nm).

It's true that the roughness within a sample is not large, but the reported range is between different samples. We have added an additional cross-sectional SEM to SI Figure S5 and have clarified the text:

The film thickness is 150 ± 43 nm (where the range arises from sample-to-sample variation) determined by cross sections of the films imaged by SEM (SI Figure S5).

From your experiments seems that NZr(NDI) MOF is an electroactive material that present a better performance than when you use the bare ligand, even when the redox mechanism is the same in both cases. Is this true? In this case, which is the Zr contribution? Participate in the charge transfer processes? Is Zr playing a role in redox processes with electron acceptors?

The Zr(NDI) film does have improved photoelectrochemical performance as compared to the bare ligand. We attribute this to enhanced charge transfer across the interface of the semiconductor when the monolayer of ligand is extended to a 3-D material with a larger number of charge acceptors AND the opportunity for spatial separation of charge carriers. We believe that the Zr nodes solely play a coordinative and structural role in the film, without participating in charge transfer itself. Evidence for this in the literature and in our own previous work shows that the Zr(IV)/(III) couple is outside the redox window that we are operating within (Gao et al, Journal of The Electrochemical Society, 166 (6) B328-B335 (2019), Mijangos et al, Dalton Trans., 2017, 46, 4907) and therefore is unlikely to be reduced.

The characterization of the films after reaction (XPS and XRD) must be included to evaluate the stability of this systems. In the case of XPS, is necessary to include the high resolution spectra.

As discussed above, unfortunately XRD could not be performed. However, we have included high-resolution XPS spectra for each sample after photoelectrochemical operation (Figure S4) showing the clear presence of the elements composing the MOF after photoelectrochemical measurements.

Is clear that light play a role in the process, but is very surprising that the redox events are not controlled by the flux. More experiments must be performed (even to lower light intensity) a and without light in order to assure this behavior and to understand better the light induced charge processes in you system.

We also were initially surprised by the lack of dependence of photocurrent on light intensity! This unexpected result is what prompted us to examine what other processes may be limiting the photocurrent. The close match of the photocurrent value to the modeling done using the finite diffusion model (equation 3) provides strong evidence that the current at all light intensities (above the threshold to produce minority carriers) is limited by the diffusion of charge carriers through the MOF. We look forward to further examining this interplay of current-limiting processes in future work as we develop methods for increased control over MOF thickness.

The majority of the comments (in blue) from reviewer 4 were sufficiently addressed (in black) and there is no need for comment on my end. I have found the few comments where I could envision there being dispute and have given my thoughts (in red) as to whether or not they are satisfactory.

X-Ray diffractogram of the film must be include to assure the purity of the MOF and discard the appearance of other phases.

For over a year we attempted to obtain PXRD data for these films. These attempts included long-scan measurements (more than 70 seconds/step) as well as grazing incidence XRD. Unfortunately, in no cases did we obtain satisfactory results. In contrast, the higher resolution SEM images again and again (Figure 1C from manuscript below as an example) showed the classic octahedral UiO-type morphology as expected for this NDI MOF (based on our prior work, see Figure 2B of JACS 2018, 140, 2985):

Also, all other characterization data (XPS, ATR-FTIR, cyclic voltammetry) support successful formation of the MOF. We ascribe the inability to obtain unambiguous XRD data mainly to two factors; the low film thickness of around 150 nm and a preferred orientation of the MOF thin film along an axis that gives weak reflections. These explanations are supported by currently unpublished data from the group where thicker films could be grown by an alternative method. Films prepared under these conditions are $> 1 \mu\text{m}$ thick (as determined by SEM), and start giving XRD patterns. We include the unpublished XRD herein for the reviewer:

To clarify this to the reader, we have added the following text to the manuscript: Extensive efforts to characterize the thin films by X-ray diffraction (XRD) either by grazing incidence XRD or with long-scan measurements (70 seconds/step) were unfortunately unsuccessful, a fact that we ascribe to the thinness of the films and their preferred orientation that precludes observation of X-ray diffraction

The response given is reasonable and from my experience with thin-film MOFs and especially since the MOF is unluckily oriented with the strongest reflection, it is not surprising to see a lack of XRD patterns taken with a standard instrument. There is a chance that one could obtain diffraction information with synchrotron-based GIWAXS measurements but those are out of the scope for this revision. Given all of the supplementary characterization provided by the authors, this response and addition to the manuscript is sufficient for me.

From your experiments seems that NZr(NDI) MOF is an electroactive material that present a better performance than when you use the bare ligand, even when the redox mechanism is the same in both cases. Is this true? In this case, which is the Zr contribution? Participate in the charge transfer processes? Is Zr playing a role in redox processes with electron acceptors?

The Zr(NDI) film does have improved photoelectrochemical performance as compared to the bare ligand. We attribute this to enhanced charge transfer across the interface of the semiconductor when the monolayer of ligand is extended to a 3-D material with a larger number of charge acceptors AND the opportunity for spatial separation of charge carriers. We believe that the Zr nodes solely play a coordinative and structural role in the film, without participating in charge transfer itself. Evidence for this in the literature and in our own previous work shows that the Zr(IV)/(III) couple is outside the redox window that we are operating within (Gao et al, Journal of The Electrochemical Society, 166 (6) B328-B335 (2019), Mijangos et al, Dalton Trans., 2017, 46, 4907) and therefore is unlikely to be reduced.

The response is sufficient for me. I would ask for this interpretation/discussion to be added to the main text.

The characterization of the films after reaction (XPS and XRD) must be included to evaluate the stability of this systems. In the case of XPS, is necessary to include the high resolution spectra.

As discussed above, unfortunately XRD could not be performed. However, we have included high-resolution XPS spectra for each sample after photoelectrochemical operation (Figure S4) showing the clear presence of the elements composing the MOF after photoelectrochemical measurements.

XPS would show that the elements within the MOF are in the same oxidation state and chemical environment before and after the test so this is a good measurement. One thing that XPS would not show very well is if there is some leeching of the MOF into the solution. Can this be tested with something like ICP of the electrolyte for the presence of the elements within the MOF? Or perhaps (photo) electrochemically by investigating the shape of the CV curves the integrated charge corresponding the reduction/oxidation of the MOF?

Is clear that light play a role in the process, but is very surprising that the redox events are not controlled by the flux. More experiments must be performed (even to lower light intensity) a and without light in order to assure this behavior and to understand better the light induced charge processes in you system.

We also were initially surprised by the lack of dependence of photocurrent on light intensity! This unexpected result is what prompted us to examine what other processes may be limiting the photocurrent. The close match of the photocurrent value to the modeling done using the finite diffusion model (equation 3) provides strong evidence that the current at all light intensities (above the threshold to produce minority carriers) is limited by the diffusion of charge carriers through the MOF. We look forward to further examining this interplay of current-limiting processes in future work as we develop methods for increased control over MOF thickness.

This is very interesting indeed. However, I would ask the authors to go even lower with light intensity to probe at which point the current does begin to decrease (they only tested 50 and 100 mA/cm² intensities). This is a very straightforward experiment and light intensities down to 1 mA/cm² should be tested as the data from and the interpretation of this experiment could be a very useful addition to improve the community's understanding of these systems.

Our responses to Referee 5 follow the comments in italics.

Referee 5:

The majority of the comments (in blue) from reviewer 4 were sufficiently addressed (in black) and there is no need for comment on my end. I have found the few comments where I could envision there being dispute and have given my thoughts (in red) as to whether or not they are satisfactory.

We thank the reviewer for their kind assessment of our responses.

X-Ray diffractogram of the film must be include to assure the purity of the MOF and discard the appearance of other phases.

For over a year we attempted to obtain PXRD data for these films. These attempts included long-scan measurements (more than 70 seconds/step) as well as grazing incidence XRD. Unfortunately, in no cases did we obtain satisfactory results. In contrast, the higher resolution SEM images again and again (Figure 1C from manuscript below as an example) showed the classic octahedral UiO-type morphology as expected for this NDI MOF (based on our prior work, see Figure 2B of JACS 2018, 140, 2985):

Also, all other characterization data (XPS, ATR-FTIR, cyclic voltammetry) support successful formation of the MOF. We ascribe the inability to obtain unambiguous XRD data mainly to two factors; the low film thickness of around 150 nm and a preferred orientation of the MOF thin film along an axis that gives weak reflections. These explanations are supported by currently unpublished data from the group where thicker films could be grown by an alternative method. Films prepared under these conditions are > 1 μm thick (as determined by SEM), and start giving XRD patterns. We include the unpublished XRD herein for the reviewer:

To clarify this to the reader, we have added the following text to the manuscript: Extensive efforts to characterize the thin films by X-ray diffraction (XRD) either by grazing incidence XRD or with long-scan measurements (70 seconds/step) were unfortunately unsuccessful, a fact that we ascribe to the thinness of the films and their preferred orientation that precludes observation of X-ray diffraction

The response given is reasonable and from my experience with thin-film MOFs and especially since the MOF is unluckily oriented with the strongest reflection, it is not surprising to see a lack of XRD patterns taken with a standard instrument. There is a chance that one could obtain diffraction information with synchrotron-based GIWAXS measurements but those are out of the scope for this revision. Given all of the supplementary characterization provided by the authors, this response and addition to the manuscript is sufficient for me.

We thank the reviewer for sharing their expertise in this positive evaluation of our efforts.

From your experiments seems that NZr(NDI) MOF is an electroactive material that present a better performance than when you use the bare ligand, even when the redox mechanism is the same in both cases. Is this true? In this case, which is the Zr contribution? Participate in the charge transfer processes? Is Zr playing a role in redox processes with electron acceptors?

The Zr(NDI) film does have improved photoelectrochemical performance as compared to the bare ligand. We attribute this to enhanced charge transfer across the interface of the semiconductor when the monolayer of ligand is extended to a 3-D material with a large number of charge acceptors AND the opportunity for spatial separation of charge carriers. We believe that the Zr nodes solely play a coordinative and structural role in the film, without participating in charge transfer itself. Evidence for this in the literature and in our own previous work shows that the Zr(IV)/(III) couple is outside the redox window that we are operating within (Gao et al, Journal of The Electrochemical Society, 66 (6) B328-B335 (2019), Mijangos et al, Dalton Trans., 2017, 46, 4907) and therefore is unlikely to be reduced.

The response is sufficient for me. I would ask for this interpretation/discussion to be added to the main text.

We have added the following as a footnote to the manuscript (ref. 34): The Zr nodes solely play a coordinative and structural role in the film, without participating in charge transfer itself, as the Zr(IV)/(III) couple is outside the redox window that we are operating within.

The characterization of the films after reaction (XPS and XRD) must be included to evaluate the stability of this systems. In the case of XPS, is necessary to include the high resolution spectra.

As discussed above, unfortunately XRD could not be performed. However, we have included high- resolution XPS spectra for each sample after photoelectrochemical operation (Figure S4) showing the clear presence of the elements composing the MOF after photoelectrochemical measurements.

XPS would show that the elements within the MOF are in the same oxidation state and chemical environment before and after the test so this is a good measurement. One thing that XPS would not show very well is if there is some leeching of the MOF into the solution. Can this be tested with something like ICP of the electrolyte for the presence of the elements within the MOF? Or perhaps (photo) electrochemically by investigating the shape of the CV curves the integrated charge corresponding the reduction/oxidation of the MOF?

We have included evidence by UV-Vis spectroscopy that there is no detectable leaching of the NDI linker into the electrolyte after PEC operation (Supplementary Figure 4).

Is clear that light play a role in the process, but is very surprising that the redox events are not controlled by the flux. More experiments must be performed (even to lower light intensity) a and without light in order to assure this behavior and to understand better the light induced charge processes in you system.

We also were initially surprised by the lack of dependence of photocurrent on light intensity! This unexpected result is what prompted us to examine what other processes may be limiting the photocurrent. The close match of the photocurrent value to the modeling done using the finite diffusion model (equation 3) provides strong evidence that the current at all light

intensities (above the threshold to produce minority carriers) is limited by the diffusion of charge carriers through the MOF. We look forward to further examining this interplay of current-limiting processes in future work as we develop methods for increased control over MOF thickness.

This is very interesting indeed. However, I would ask the authors to go even lower with light intensity to probe at which point the current does begin to decrease (they only tested 50 and 100 mA/cm² intensities). This is a very straightforward experiment and light intensities down to 1 mA/cm² should be tested as the data from and the interpretation of this experiment could be a very useful addition to improve the community's understanding of these systems.

We have included data probing the film's photoelectrochemical response down to 10 mW cm⁻², under which intensities the current is still constant, indicating a diffusion-controlled response even at low light intensities (Supplementary Fig. 19). We agree that this is quite intriguing in a photoelectrochemical system and we are further exploring this in follow-up work.